# Demonstration of Ru as the 4th ferromagnetic element at room temperature

P. Quarterman [1], Congli Sun[2], Javier Garcia-Barriocanal[3], Mahendra DC[4], Yang Lv [1], Sasikanth Manipatruni[5], Dmitri E. Nikonov [5], Ian A. Young[5], Paul M. Voyles [2] & Jian-Ping Wang[1]

Development of novel magnetic materials is of interest for fundamental studies and applications such as spintronics, permanent magnetics, and sensors. We report on the first experimental realization of single element ferromagnetism, since Fe, Co, and Ni, in metastable tetragonal Ru, which has been predicted. Body-centered tetragonal Ru phase is realized by use of strain via seed layer engineering. X-ray diffraction and electron microscopy confirm the epitaxial mechanism to obtain tetragonal phase Ru. We observed a saturation magnetization of 148 and 160 emu cm$^{-3}$ at room temperature and 10 K, respectively. Control samples ensure the ferromagnetism we report on is from tetragonal Ru and not from magnetic contamination. The effect of thickness on the magnetic properties is also studied, and it is observed that increasing thickness results in strain relaxation, and thus diluting the magnetization. Anomalous Hall measurements are used to confirm its ferromagnetic behavior.

[1] Department of Electrical and Computer Engineering, University of Minnesota, Minneapolis, MN 55455, USA. [2] Department of Materials Science and Engineering, University of Wisconsin, Madison, WI 53706, USA. [3] Characterization Facility, University of Minnesota, Minneapolis, MN 55455, USA. [4] School of Physics and Astronomy, University of Minnesota, Minneapolis, MN 55455, USA. [5] Components Research, Intel Corp., Hillsboro, OR 97124, USA. Correspondence and requests for materials should be addressed to J.-P.W. (email: jpwang@umn.edu)

Ferromagnetism has a long history of study, and to date only three single elements display room temperature ferromagnetism—Fe, Co, and Ni[1]. Gd nearly misses room temperature ferromagnetism with a Curie temperature of 293 K[2], and Pd and Pt narrowly miss the Stoner criteria[3]. In addition, countless alloys have been produced, which are composed of at least one 3d element, such as NiFe, FePt, CoPt, $SmCo_5$, MnBi, and more; they vary in anisotropy constant, $K_u$, from $10^5$ to $10^8$ erg $cm^{-3}$. Magnetic thin film materials with large magnetocrystalline anisotropy (MCA) are desired for their technological applications such as spin torque transfer random access memory (STT-RAM), heat assisted magnetic recording (HAMR)[4], and more. Large MCA in magnetic materials is due to the involvement of 4d and 5d valence shells, which makes magnetism in metals with these valence shells of great interest. Bulk Ru is paramagnetic, stable in a hexagonal close pack (HCP) crystal structure, and has a long history of use in as a seed layer in hard disk media[5], a spacer in synthetic antiferromagnetic structures that are used for both magnetic read heads and STT-RAM[6], and also can be used as a capping layer due to its chemical stability. Theoretical calculations have proposed ferromagnetism is possible in Ru, Os, and Ur if these elements can be forced into a tetragonal lattice structure[7–9]. Shiki et al.[10] claimed to have formed body-centered tetragonal (BCT) Ru by growing on single crystal (110) Mo substrates, but they did not report the magnetic properties of their thin films. More recently it has been proposed that tetragonal Ru may have large perpendicular MCA, up to two orders of magnitude greater than traditional 3d magnetic metals due to Jahn–Teller splitting[11]. Due to its inherit metastability, BCT Ru cannot be grown in bulk. However, by utilizing strain in thin films it is possible to realize a metastable phase given properly lattice matched seed layers to induce nucleation of this metastable BCT phase in ultra-thin Ru films. A BCT structure with lattice constant $a = 3.25$ Å and distorted such that $c/a = 0.84$, has been predicted to have large MCA[11].

In this work, we synthesize ultra-thin BCT Ru using epitaxial growth on seed layers with appropriate lattice matching. The epitaxial relation to grow BCT Ru is understood using X-ray characterization and electron microscopy. Ferromagnetism from the tetragonal Ru is measured using magnetometry and transport methods. We observe a magnetization of 148 and 318 emu $cm^{-3}$ from magnetometry and transport measurements, respectively. This discrepancy is due to error in knowing the precise Ru magnetic volume when calculating the magnetization from the measured magnetic moment. Additionally, appropriate control measurements for each of the mentioned techniques are employed in this work to rule out magnetic contaminants.

## Results

**Crystallography of samples.** Epitaxial Ru films with nominal thickness of 2.5, 6, and 12 nm were grown using a sputtering system on $(11\bar{2}0)$ $Al_2O_3$ substrates with a 20 nm (110) Mo seed layer at 400 °C. A non-textured stack with a 20 nm Mo seed layer and 2.5 nm of Ru was grown at room temperature as a control sample, and a 20 nm (110) Mo layer, with no Ru, sample was grown at 400 °C on a $(11\bar{2}0)$ $Al_2O_3$ substrate as an additional control. (110) Mo lattice matches and grows well on $(11\bar{2}0)$ $Al_2O_3$-oriented substrates at an offset angle of ~35°[12, 13]. The crystallographic families of planes $(11\bar{2}0)$ and (0001) of the $Al_2O_3$ substrate can be represented as (110) and (001), respectively, and will be denoted as such throughout the remainder of this paper. A cartoon of the expected epitaxial relationship for the film stack of (110) $Al_2O_3$//(110) Mo//$(01\bar{1})$ Ru is shown in Fig. 1a. This epitaxial relationship has been confirmed using X-ray diffraction (XRD) in the grazing incidence configuration, and by rotating the sample 360°. As shown in Fig. 1b, the expected 4-fold symmetry for the (110) planes of $Al_2O_3$ has been observed, and the

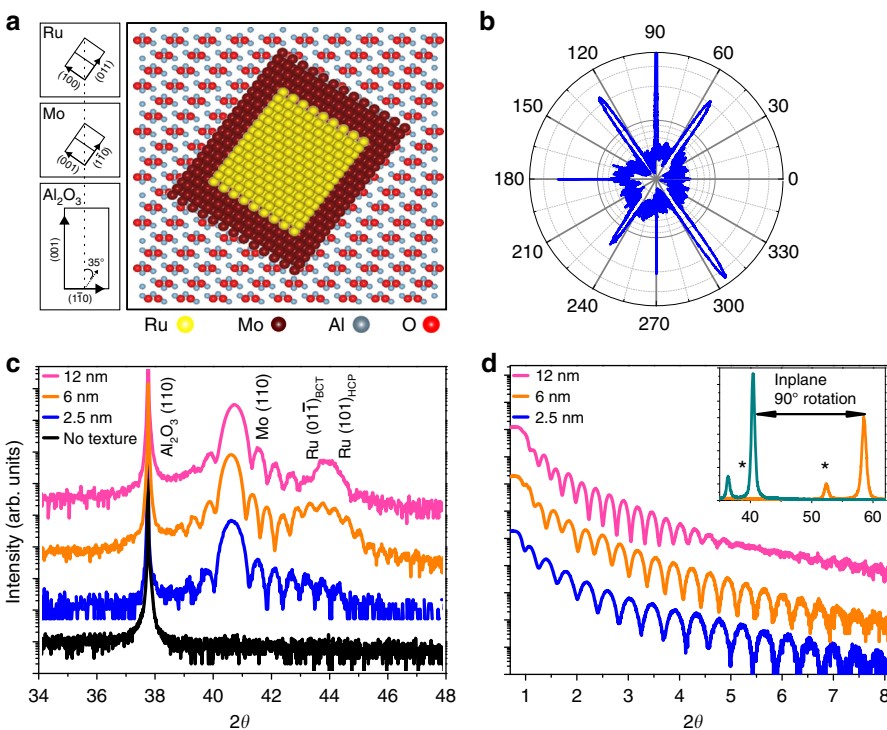

**Fig. 1** Epitaxial relation for tetragonal Ru growth. The expected crystallographic structure and epitaxial relation is shown using **a** a cartoon showing the epitaxial growth relation for (110) $Al_2O_3$//(110) Mo//$01\bar{1}$Ru, and **b** X-ray diffraction using grazing incidence, by placing the detector in the (110) plane of $Al_2O_3$ and rotating the sample over 360°. **c** Conventional XRD spectra with the orientation parallel to {110} $Al_2O_3$, and **d** X-ray reflectivity for 2.5, 6, and 12 nm Ru films; with the inset of **d** showing the grazing incidence coupled scan when aligned to the short (black) and long (blue) edge of (110) Mo

predicted Mo crystallographic orientation is rotated approximately 35° from the (001) substrate plane. The coupled $\theta$–$2\theta$ XRD scans, for all four samples, are shown in Fig. 1c. The sample grown at room temperature does not show any evidence of texturing in the thin films, but the samples with Ru layer with nominal thickness of 2.5, 6, and 12 nm grown at 400 °C, show highly textured (110) Mo. The scans show finite size diffraction fringes corresponding to the total thickness of the sample, and the low roughness of the interfaces. A peak, which corresponds to both the $(01\bar{1})$ BCT phase Ru at 43.25° or (101) HCP phase Ru at 44.02°, increases in intensity and shifts from 43.31° to 43.96° from 2.5 to 12 nm sample. Since with increasing Ru thickness, the peak shifts to larger $2\theta$ angle, narrows in width, and the finite thickness fringes disappear, XRD scans suggest a mixing of BCT and HCP phases is present, and the HCP to BCT ratio increases with Ru thickness. This two-phase nature of the films is further supported by cross-section electron microscopy—discussed later in this paper. In addition, the desired BCT Ru structure, with $a = 3.25$ Å, is strained to a smaller interatomic spacing than is predicted. This is because the (001) Mo plane (3.15 Å) is straining the (100) BCT Ru plane. Such a strain on the BCT Ru lattice would shrink $a$ to <3.25 Å, which creates a distortion of the BCT Ru phase, and will increase $2\theta$ for the $(01\bar{1})$ BCT Ru plane. X-ray reflectivity (XRR) for the textured samples is shown in Fig. 1d, and the oscillations to 10° show the low roughness of the interfaces of the samples. The reflectivity curve was fit using GenX[14], which confirmed the film thicknesses for the Mo and Ru thin films, and a low interface roughness of less than 3 Å. The in-plane lattice parameters for the Mo layer were measured using the grazing incidence XRD configuration, as shown in the inset of Fig. 1d, and Mo lattice constants matching the theoretical values were observed. The roughness of the Ru surface was found to be 0.21, 0.13, and 0.21 nm for 2.5, 6, and 12 nm Ru films, respectively, as measured by atomic force microscopy (AFM), shown in Supplementary Fig. 1, which closely matches the results of the XRR fits.

Cross-section STEM images along the [001] $Al_2O_3$ zone axis, which matches the [111] Ru zone axis, of the 6 nm Ru sample, were obtained, and are shown in Fig. 2. The high-angle dark field (HAADF) STEM image (Fig. 2b) shows that the Ru and Mo layers are highly textured, and the tetragonal structure of the Ru layer has been confirmed using high-precision HAADF STEM[15]. There are also clear distortions in the Ru epitaxy, which can be viewed as a shift of every other $(1\bar{1}0)$ plane, as shown in the high-precision HAADF STEM image. This distortion is believed to be caused by the mismatch between (001) Mo and (100) Ru, as pointed out in the XRD discussion earlier. The grain boundary shown in Fig. 2c is formed by different orientations of the lattice distortion, and two equivalent $\{10\bar{1}\}$ surfaces of tetragonal Ru on the [111] zone axis. Fast Fourier transform (FFT) of the STEM (red boxed region of Fig. 2b) has been used to determine that the structure of the Ru film is distorted BCT (Fig. 3). A cartoon to visualize the expected FFT pattern along the [111] zone axis for tetragonal Ru is shown in Fig. 3b. The lattice parameters, $c$ and $a$, were estimated using fitting; it was completed with the assumption that the crystal is a perfect tetragonal structure, and using the measured interatomic distances ($d_1$, $d_2$, $d_3$) and atomic plane angles ($\theta_1$, $\theta_2$, $\theta_3$). Using this method, $d_1/d_2$ has been calculated and leads to a $c/a$ ratio of 0.88, which corresponds to $a = 3.19 \pm 0.3$ Å, and is near the optimal prediction for BCT Ru of 3.25 Å; furthermore, it is not surprising to see minor deviations given that the Ru film has distortions in the tetragonal structure, which cannot be accounted for in this model. In addition, cross-section STEM images were also collected on 2.5 and 12 nm Ru samples (Supplementary Fig. 2). These STEM images show that at 2.5 nm, the Ru thin film is predominately in the BCT phase. In the 12 nm Ru sample,

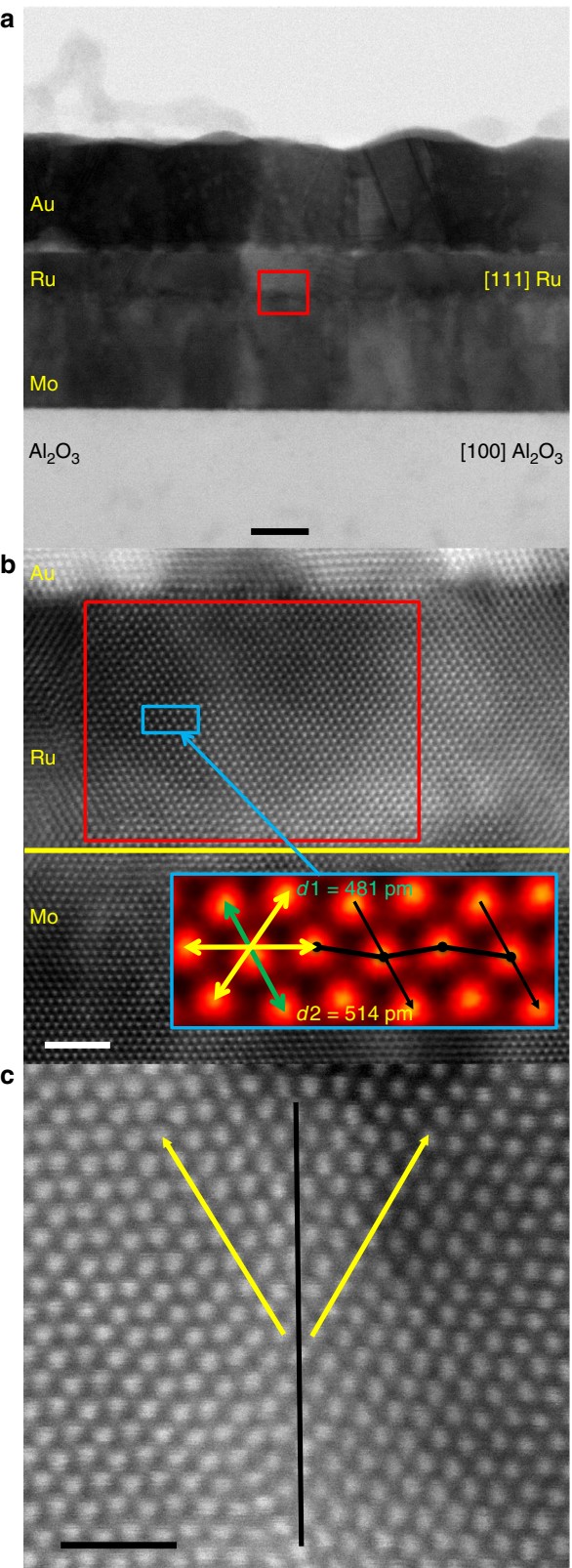

**Fig. 2** Characterization of Ru tetragonal structure. Cross-section STEM images along the [001] zone axis of $Al_2O_3$. **a** The annular bright field (ABF) STEM images of the full sample stack. Scale bar is 10 nm. **b** High-angle annular dark field (HAADF) STEM at the Mo–Ru interface, with the inset showing the high-precision HADDF STEM for the [111] Ru zone axis using the non-rigid registration method[15]. The lattice distortion is shown using the dashed black line. Scale bar is 2 nm. **c** A HAAADF STEM zoom in on a Ru grain boundary due to equivalent surfaces growth surfaces. Scale bar is 1 nm

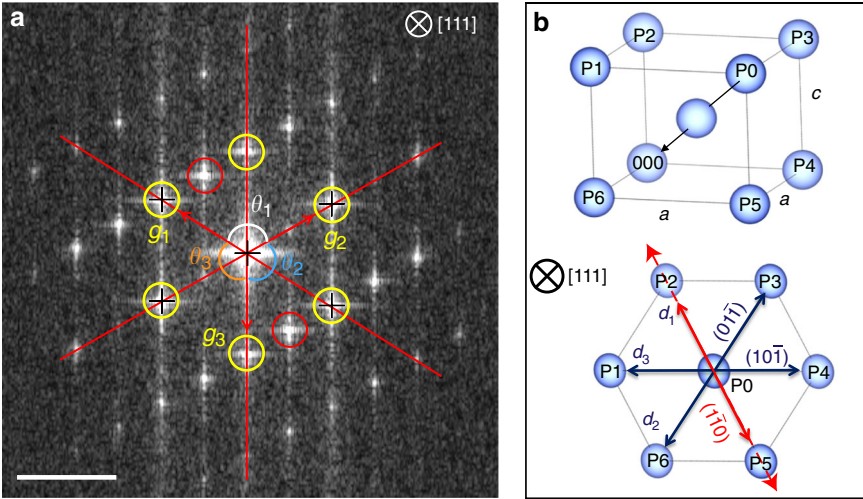

**Fig. 3** Determination of Ru lattice parameters. **a** FFT of the 6 nm Ru film along the [111] zone axis, from indicated region of Fig. 2b, with the expected BCT grouping highlighted in yellow, and distortions in the tetragonal ordering are highlighted in red. Scale bar is 5 nm$^{-1}$. **b** The expected [111] zone axis projection for BCT Ru, and by using the measured interatomic spacing and atomic plane angles, an estimate of $c/a$ for a BCT structure can be calculated. The dashed red line shows the orientation of the lattice distortion

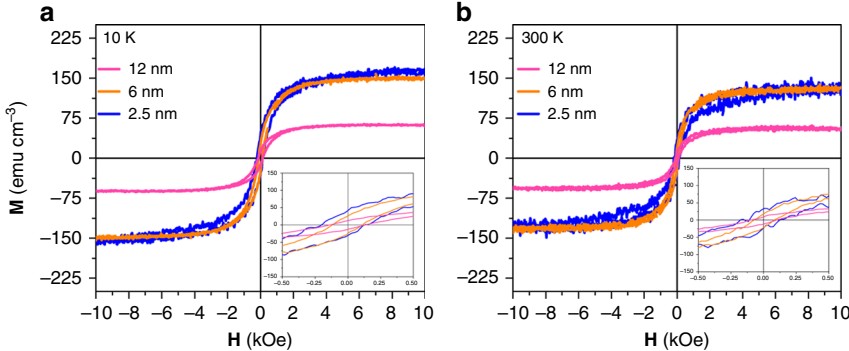

**Fig. 4** Observation of ferromagnetism. Magnetization vs. field hysteresis curves with an in-plane field orientation for 2.5, 6, and 12 nm of Ru at **a** 10 K and **b** room temperature, with the inset for each showing a zoom in near the coercive field region

while there are clear regions of BCT Ru spanning the entire film thickness, there are also regions with two competing phases of BCT and HCP—which supports to XRD data discussed earlier. Given that STEM images were collected at a zone axis correlating to the BCT Ru structure, specific HCP Ru crystallographic information cannot be determined from these images. It should also be noted that two-phase regions, as shown in 12 nm Ru films of Supplementary Fig. 2, exist in 2.5 and 6 nm films, but it is clear that two-phase Ru regions increase in frequency as the Ru film thickness increases.

**Magnetic characterization.** The in-plane magnetization vs. field (**M**–**H**) hysteresis loops for 2.5, 6, and 12 nm Ru films, grown at high temperature, were measured at 10 and 300 K (Fig. 4) by a vibrating sample magnetometer (VSM). The **M**–**H** loops display clear ferromagnetic behavior with a maximum $M_s$ of 160 and 148 emu cm$^{-3}$ at 10 and 300 K, respectively, in the 2.5 nm thick Ru film. $M_s$ was calculated by assuming the entire Ru layer is magnetic, and $M_s$ was found to decrease with increasing Ru thickness. Multiple samples, and multiple **M**–**H** curves were measured, for both the samples, and are shown in Supplementary Table 1 in the supplementary materials. Given that 2.5 and 6 nm Ru samples show a similar room temperature $M_s$, a mean $M_s$ value of 141 emu cm$^{-3}$ was calculated by averaging $M_s$ from all samples with

these thicknesses, which assists in allaying contamination concerns. All samples show a coercivity ($H_c$) of ~130 Oe. In order to be further certain that the observed ferromagnetism is not due to sample contamination, the same sample holder used for the Ru measurement was measured alone (after each sample), at 10 and 300 K, and only a paramagnetic signal was measured Supplementary Fig. 2a is such an example). Furthermore, the measurements were repeated several times with multiple holders to ensure results are not an artifact. To further support BCT textured Ru, not magnetic contamination, is responsible for the ferromagnetic **M**–**H** loop measurements on the room temperature deposited samples with no crystallographic texture were collected, and none display ferromagnetism (Supplementary Fig. 2b is an example, which is repeatable). A textured (110) Mo sample grown on (110) Al$_2$O$_3$ at 400 °C was also measured (no Ru layer on top) as an additional control, and also showed no ferromagnetic behavior (Supplementary Fig. 2b); this rules out ferromagnetism from the Mo layer, and any possible magnetic contamination introduced from in situ heating of the samples during deposition.

Room-temperature transport measurements were collected on the 6 nm textured BCT Ru sample, with the same (110) Al$_2$O$_3$ substrate and 20 nm (110) Mo seed layer, by measuring the Hall resistance (**R**$_{Hall}$) as a function of external field ($H_z$) using the

Vander Pauw method. A dc current for Hall measurements of 0.1 mA was used. $\mathbf{R}_{Hall}$ increases linearly as external out-of-plane field is increased, and it saturates when all in-plane moments are pulled out-of-plane near ±4 kOe as shown in Fig. 5. In ferromagnets, the Hall resistance is given by $\mathbf{R}_{Hall} = \mathbf{R}_o + \mathbf{R}_{AHE}$, where $\mathbf{R}_o$ is the ordinary Hall resistance and proportional to the external field; $\mathbf{R}_{AHE}$ is the anomalous Hall resistance and is proportional to the perpendicular component of magnetization[16–18]. At lower external field $\mathbf{R}_{AHE}$ is dominant whereas at larger field $\mathbf{R}_o$ dominates. The saturation external field is equivalent to the demagnetization field ($4\pi M_s$ for a thin film), which corresponds to the $M_s = 318$ emu cm$^{-3}$. Another control sample with a stack structure of substrate/Mo(20)/Ru(6), which has been shown to have no texture in the Mo or Ru layers (Fig. 1c), was measured and only shows a linear field dependent resistance due to the ordinary Hall effect. This supports our claim that the observed magnetism is due to Ru BCT texturing, and not from any magnetic contamination. Furthermore, the transport measurement rules out the possibility of the magnetism observed coming from dust, since dust cannot contribute to the conduction. A maximum field of 5.5 kOe along the forward sweep direction was used since 5.5 kOe is enough to show the relevant anomalous and ordinary Hall effects, and any additional field just adds more of the ordinary Hall effect dominated region. Since the BCT Ru does not have any perpendicular anisotropy, no anisotropy in $\mathbf{R}_{Hall}$ is expected; so the forward sweep is enough to simply confirm the ferromagnetic behavior of the sample and as an additional estimate of $M_s$.

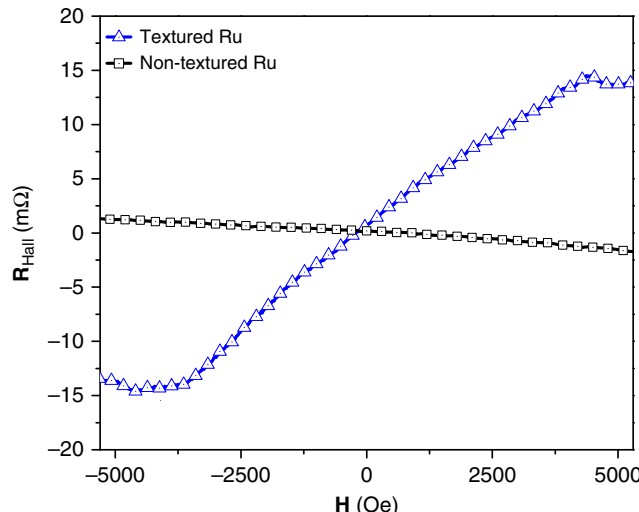

**Fig. 5** Hall resistance confirmation of ferromagnetism. Hall resistance ($\mathbf{R}_{Hall}$) vs. **H** (applied perpendicular to surface) for textured (blue) and non-textured (black) Mo/Ru films. The substrate/Mo/Ru sample, which has no crystallographic texture, shows only the ordinary Hall effect, but the Ru sample with BCT texture shows the anomalous Hall effect in addition to the ordinary Hall effect. Given that the Ru samples do not have a perpendicular easy axis, the resistance will change once the field is strong enough to saturate the demagnetization field of $4\pi M_s$, corresponding to a $M_s$ of ~318 emu cm$^{-3}$. The saturated regions are designated by black arrows

## Discussion

Odkhuu et al.[11] predicted a $M_s$ of ~400 emu cm$^{-3}$, and a perpendicular $K_u$ of ~$10^7$ erg cm$^{-3}$, if the optimal tetragonal lattice is realized with the (001) tetragonal Ru orientation being out-of-plane. The measured $M_s$ of 160 emu cm$^{-3}$, at 10 K by VSM, is lower than the predicted 400 emu cm$^{-3}$, and decreased further with increased Ru thickness. $M_s$ was calculated assuming the entire volume of the Ru layer is uniformly ferromagnetic, which has been shown to not be an accurate assumption, and is the major source of error in precisely calculating $M_s$. For example, it is clear from XRD spectra and STEM images that some Ru grains in the samples do not have BCT structure, and so this is a magnetic "dead" region that dilutes the calculated magnetization. The frequency of these non-BCT Ru grains increases with film thickness, and is the reason for the dependence of $M_s$ on Ru thickness. The $\mathbf{R}_{Hall}$–**H** measurement (Fig. 5) shows a $M_s$ closer to the prediction, of ~318 emu cm$^{-3}$, which is not surprising since this technique does not rely on precisely knowing the volume of ferromagnetic Ru. Additionally, the lower than expected $M_s$, from both VSM and AHE measurements, can be due to both the $c/a$ ratio not being exactly 0.84, as predicted, and the observed lattice distortions. Furthermore, it is not yet clear how the distortion of the tetragonal lattice affects the magnetic properties. The **M**–**H** measurements do not show a magnetic easy axis, which is to be expected since the (001) BCT Ru axis does not align perpendicular to the substrate. The low coercivity and remanence observed can be explained by the multiple available equivalent growth orientations, the continuous thin films being in the free domain wall limit, and mismatched tetragonal grains for the in-plane orientation.

In summary, we have developed a room temperature ferromagnetic metastable tetragonal phase of Ru by using a Mo seed layer to strain the Ru thin film. Our $c/a$ ratio of 0.88 was found to be within error of the predicted ratio of 0.84. Using textured and non-textured seed layers, we have confirmed the observed ferromagnetism is not due to contamination in the sample. A

magnetization of 148 emu cm$^{-3}$ was found at room temperature, and using multiple samples and **M**–**H** measurements, a mean value for the magnetization of 141 emu cm$^{-3}$ was calculated. The metastable tetragonal structure of the Ru layer has been confirmed, and distortions were observed, which may dilute the magnetization and MCA. The (001) tetragonal Ru plane does not lie perpendicular to the substrates which leads to a soft coercive field, however, if out-of-plane texturing can be achieved high $K_u$ Ru may be realized. The thickness dependence was also examined, and it has been found that due to Ru relaxing into a non-ferromagnetic phase, the magnetization drops with increasing thickness. Transport measurements of the anomalous Hall resistance supported the observation of ferromagnetism from BCT textured Ru. This work experimentally demonstrates that it is possible to create single element ferromagnetic materials at room temperature, based on Jahn–Teller magnetization, outside of Fe, Co, and Ni.

## Methods

**Thin film sample preparation**. Ru thin films with a thickness of 2.5, 6, and 12 nm, were grown on Al$_2$O$_3$ substrates cut along the (11$\bar{2}$0) orientation, also known as (110), with a 20 nm Mo seed layer. The thin films were grown using a eight-target UHV sputtering system with base pressure of $8 \times 10^{-8}$ Torr or lower; with one 2.5 nm Ru sample grown at room temperature, and another 2.5 nm, as well as 6 and 12 nm, heated to 400 C in situ with a 1 h post anneal immediately after deposition.

**Crystallographic characterization**. The crystallographic information was collected using conventional and grazing incidence XRD, and XRR with a Panalytic X'pert Pro (Cu-Kα) system. Cross-section STEM samples were prepared by in situ lift out using a Zeiss Auriga focused ion beam (FIB), the samples were coated with Au in order to protect the Ru surface during preparation. The final FIB milling voltage is dropped to 2 kV to minimize damage from implanted Ga. STEM imaging is performed on an FEI Titan with CEOS probe aberration corrector operated at 200 kV with a probe convergence angle of 24.5 mrad, spatial resolution of 0.08 nm, probe current of ~20 pA and by stabilizing the sample for 6 + hours inside the TEM column, with 20 µs pixel dwell time.

**Magnetic measurements**. Magnetization vs. field (**M**–**H**) loops were measured using a VSM at 10 and 300 K, respectively. Transport measurements to examine

the anomalous Hall effect were conducted using a Quantum Design Physical Property Measurement System with magnetic field applied perpendicular to as-deposited 5 × 5 mm samples with the Vander Pauw configuration. In addition to testing a BCT Ru textured sample, a non-textured with same stack structure was measured under the same conditions. AFM was acquired using a Bruker Nanoscope V Multimode 8.

**Data availability**. That data that support the findings of this study are available from the authors on reasonable request; see author contributions for specific data sets.

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

## Acknowledgements

The authors would like to thank Drs. Ramamoorthy Ramesh from UC Berkeley, Weigang Wang from Arizona State University, Delin Zhang from the University of Minnesota, and Julie Borchers, Michelle Jamer, and Brian Kirby from NIST for invaluable discussion. This work was supported by C-SPIN, one of six STARnet program research centers, and by the University of Minnesota Distinguished Doctoral Fellowship. Parts of this work were carried out in the Characterization Facility, University of Minnesota, which receives partial support from NSF through the MRSEC program, and the Materials Science Center, University of Wisconsin-Madison, which is supported by the NSF MRSEC program (DMR-1720415).

## Author contributions

P.Q. and J.-P.W. designed the experiment. P.Q. grew the films, measured the *M–H* curves, X-ray characterization, worked out the epitaxial relationship, analyzed the additional characterization into this finished work and wrote the manuscript. C.S. and P. M.V. completed the cross-section STEM measurements, and assisted with in-depth analysis of understanding the Ru structure. J.G.-B. contributed with XRD and XRR measurements, as well as understanding crystallography, modeling the system, and provided invaluable discussion to improve this work. M.D.C. carried out transport measurements, and worked closely on the analysis of these measurements. Y.L. completed the AFM measurements, and the associated analysis. S.M., D.E.N., and I.A.Y. were involved with the initial suggestion for examining Ru. J.-P.W. coordinated the project and was intimately involved with discussion and analysis on all fronts, as well as preparing this manuscript.

## Additional information

**Competing interests:** The authors declare no competing interests.

