## [Peer Review File · Nature Communications]

Reviewers' comments:

Reviewer #1 (Remarks to the Author):

This paper contains a very important conclusion that the authors discovered that the tetragonal Ru shows ferromagnetism at room temperature. It is very exciting if it is correct of course. Before the authors reach this conclusion, they should be more careful. The reviewer is of the opinion that I cannot judge that the results presented in this paper show tetragonal Ru is ferromagnetic because of the following reasons.

For one thing, M-H curves are so sensitive to impurities. Hence the reviewer strongly recommends the authors to employ XMCD to detect element specific magnetic moment. By using XMCD, we can focus on the magnetic properties only of tetragonal Ru even if the film contains other contaminants. This is because the authors' group have done a lot of studies in XMCD.

For another thing, "tetragonal" is not supported by evidence. Why didn't you just measure the XRD diffractions and deduce a c/a ratio from the peak position ? It is more reliable to deduce lattice constants from XRD measurements.

Ref. 14 is about Pt not about Ru. The authors need to add explanations about why ref. 14 is cited in line 88.

Reviewer #2 (Remarks to the Author):

RE: Demonstration of the 4th ferromagnetic element at room temperature: Ru

The manuscript by Quarterman et al., claims that they have experimentally demonstrates Ru to be another (the fourth) room temperature ferromagnetic element.

The possibility of BCC or BCT structured Ru being ferromagnetic had been proposed by theory dated back to 1995, and such BCT structure was realized through interfacial engineering during epitaxy by Shiki et al. The authors in this work demonstrate that such Ru BCT structures achieved through epitaxial growth are indeed ferromagnetic, and thereby closed the loop. For this work to not be considered as incremental, the authors need to provide more substantial new understanding. In its current form, I recommend the authors submit to a more specific journal.

The authors demonstrated that it is possible to use Ru to create single element ferromagnetic materials at room temperature with a saturated magnetization of 148 emu/cm³, which can be increased to about 350 emu/cm³ after field annealing. These results are interesting and can provide additional insights for researchers in this field. To make the work more complete, I have the following suggestions:

1. In Fig. S2, the magnetic properties of epitaxial Mo/Al₂O₃ should also be shown as a reference and compared with that of the Ru/Mo/Al₂O₃ structures. The magnetic properties of Mo/Al₂O₃ are not clear in the manuscript. In addition, substrate heating during growth can be a potential source for introducing magnetic contaminations.
2. The saturated magnetization decreases with increasing the thickness of Ru. The authors argued that it is due to Ru relaxation into a non-ferromagnetic phase. More careful and strict microscopy studies may strengthen this claim, as current XRD data are not convincing. If the trend is true, more data points (varying thickness) are needed to show the critical thickness or turning point. Is there any possibility that the magnetism is resulting from Mo/Ru intermixing or interface coupling?
3. If the BCT is stabilized by interfacial strain, and is the driving force for ferromagnetism, can the authors connect the strain differences due to thermal expansion at different T with changes in ferromagnetism?
4. For ferromagnetic materials, it is always useful to determine their Curie temperature. For the 2.5 nm Ru sample, 10 kOe and 100 °C were chosen as the annealing conditions. Is 100 °C above the Curie temperature? Magnetic field, T, and film thickness should be changed to provide more insights.
5. It is difficult to visualize the changes in 2.5 nm field annealed sample. XRD and XRR results shown in Fig. S3 should be added to Fig. 1, overlapping with as-grown 2.5 nm sample.
6. Line 99, given the epitaxial relationship between Ru and Mo, the deviation should be estimated and compared to what predicted by calculation.

The manuscript still contains many typos and mistakes:

1. MCA defined twice on the first page;
2. Line 54, UHV) should be UHV;
3. Line 71, RU should be Ru;
4. Line 108, “2.5 nm thick” was changed to “3 nm thick” in Figure description.
5. Line 124, which..., which...
6. Line 145, “cubic” should be “tetragonal”?

Reviewer #3 (Remarks to the Author):

The authors claimed that they have developed a room temperature ferromagnetic metastable tetragonal phase of Ru, which indicates ferromagnetism at room temperature as the 4th single

element ferromagnetic materials at room temperature outside of Fe, Co and Ni. The work is very interesting and important for magnetic and related community. However, there are some issues unclear, and they need more experiments for publishing these results in Nature communications.

1, The magnetization measurement is a critical issue. I did not find the measurement method description in the text. Measured by SQUID? If yes, more magnetic measurements, such as magnetic spectrum or microscopic measurements, are needed to verify their claim.

2, The film of 2.5nm was annealed and its magnetization was increased dramatically, how about the other thickness samples?

3, In the abstract, it seems that they prepared 2.5, 6, and 12 nm thick samples, but in Fig4, the samples are 3, 6, 12nm. Are they right?

4, There are some typos, For example, RU, should be Ru?, ferromagnetic metastable cubic phase of Ru, should be tetragonal?

Detailed replies to the reviewer's comments:

Reviewer #1 (Remarks to the Author):

This paper contains a very important conclusion that the authors discovered that the tetragonal Ru shows ferromagnetism at room temperature. It is very exciting if it is correct of course. Before the authors reach this conclusion, they should be more careful. The reviewer is of the opinion that I cannot judge that the results presented in this paper show tetragonal Ru is ferromagnetic because of the following reasons.

Reply: Thank you very much for your review and comments. We hope we have addressed the concerns you have raised, and have replied to each of your comments, as denoted in red.

1. For one thing, M-H curves are so sensitive to impurities. Hence the reviewer strongly recommends the authors to employ XMCD to detect element specific magnetic moment. By using XMCD, we can focus on the magnetic properties only of tetragonal Ru even if the film contains other contaminants. This is because the authors' group have done a lot of studies in XMCD.

Reply: We agree that XMCD is the best option, we have been applying for beam time at the Advanced Photon Source in the middle of 2018 to look at the Ru L₃ edge for additional studies, but we are not sure when this can be completed. However, given the extensive control measurements, we believe contamination can safely be ruled out. The first control sample relies on using room temperature deposition of the exact same stack structure (substrate\Mo(20)\Ru(x), which leads to samples with the same elemental stack structure, but which have no crystallographic texture (shown in Fig 1 for x = 2.5 nm to be true). Since the measurements on this sample (and others of the same structure to repeat), we can rule out that impurities cannot be from the target source itself, and systematic errors with our M-H measurement system. The next control sample (added from prior submission, Fig S3b) is a sub/Mo(20) film grown at 400 °C, so that the Mo is highly (110 textured)—this sample showed no ferromagnetism in the M-H curves. This allows to possible sources of contamination to be ruled out: first that the Mo is the source of

ferromagnetism, and second that contamination comes from the *in situ* heating processes during deposition. Finally, we have added Table 1 into the supplemental materials to summarize the total number of each samples we have repeated, which show ferromagnetism or no ferromagnetism, and the total combined M-H curves measured for each sample. If the ferromagnetism is due to contamination, we do not expect to see such repeatable results given that we run a ‘clean’ VSM holder through our system routinely to ensure there is not contamination directly in the system. This is combined with after each Ru M-H curve is measured, the sample is removed and the same sample holder is measured independently to ensure it shows a pure dia/para-magnetic curve—any data collected that does not have such a clean VSM sample holder is discarded as a contaminated measurement.

As another independent and insensitive method to any contamination from VSM measurement, we have carried out an anomalous Hall effect measurement on a textured and non-textured (sub\Mo(20)\Ru(6) deposited at room temperature, same as discussed in prior paragraph) samples (Fig. 5 in new manuscript). The textured 6 nm Ru sample shows ordinary Hall effect in addition to AHE, whereas the non-textured sample only shows the ordinary Hall effect. This result is shown in Fig. 5 and the related discussion is on pages 10-12. It is well known that AHE signal only responds to the perpendicular magnetization component of the sample, and since the samples in this work have no easy axis, a Hall resistance change is expected at the field necessary to overcome the demagnetization field of a thin film ($4\pi M_s$). This also helps rule out any contamination from ferromagnetic dust particles, since dust will not contribute to the conduction; if dust impurities in the target or from dust are the cause of ferromagnetism they would also show up in the control sample discussed.

2. For another thing, “tetragonal” is not supported by evidence. Why didn’t you just measure the XRD diffractions and deduce a c/a ratio from the peak position? It is more reliable to deduce lattice constants from XRD measurements.

Reply: Thank you for comments, but we have initially shown the Ru structure is BCT from the cross-section analysis (Fig. 2 and Fig. 3), as detailed calculations related to the based on the use of high precision HAADF STEM cross sections, and a prior use of this method is cited in Ref. 15 (Yankovich et al) of the revised manuscript. We have added discussion on page 4 of the main manuscript concerning correlating the XRD measurements with the STEM analysis. In Cu-K α

XRD the (01 $\bar{1}$) BCT Ru plane ($2\theta = 43.25^\circ$) lies quite close to the (101) HCP Ru ($2\theta = 44.02^\circ$) plane, which is why we were hesitant to definitively claim the BCT phase based upon XRD. We have added additional STEM images of the 2.5 and 12 nm Ru films, which now supports our claim of BCT Ru, and matches with the XRD scans. In the XRD scans (Fig 1c), it is clear that at 2.5 nm there are finite thickness oscillations, which reach into a secondary peak at larger 2θ than the (110) Mo peak at approximately 43.31° , but as the thickness increases, this peak shifts to larger 2θ , and at 12 nm is near 43.96° . As the thickness increases, so does the 2θ angle, the finite thickness fringes disappear surrounding the Ru peak, combined with the peak width narrowing, and this implies the HCP phase begins to dominate the Ru film with increasing thickness. This is further supported by the new STEM images added in what is now Fig. S2 (prior Fig S2 bumped to Fig S3). STEM analysis shows that in 2.5 nm the Ru films are nearly entirely BCT Ru phase; at 12 nm Ru BCT regions are still discernable, but regions which crystal structure cannot be well determined by high resolution STEM due to overlapping phases exist. Finally, the STEM also shows that even at 12 nm nominal Ru thickness, there are BCT Ru grains than span the entire film thickness, however, at increased thickness the prevalence of mixed phase regions grows, which explains the decrease in M_s with increasing Ru thickness.

The distortion to the tetragonal lattice can be understood since BCT Ru with a of 3.25 \AA is predicted, but the (100) Ru plane grows epitaxially to the (001) Mo plane ($a = 3.15 \text{ \AA}$). Thus, the Mo layer will distort the Ru structure.

3. Ref. 14 is about Pt not about Ru. The authors need to add explanations about why ref. 14 is cited in line 88.

Reply: We wish to clarify why we cite ref. 14 by Yankovich et al (now ref 15 in revised manuscript) is cited, in what was previously line 88, since this refers to previous works which demonstrated the use of high precision HAADF STEM to understand the structure in Pt particles, which is the technique used in this work to confirm the BCT structure and lattice parameters a and c in the initial manuscript. We have added XRD discussion to further support our STEM analysis claims.

Reviewer #2 (Remarks to the Author):

The manuscript by Quarterman et al., claims that they have experimentally demonstrates Ru to be another (the fourth) room temperature ferromagnetic element.

1. The possibility of BCC or BCT structured Ru being ferromagnetic had been proposed by theory dated back to 1995, and such BCT structure was realized through interfacial engineering during epitaxy by Shiki et al. The authors in this work demonstrate that such Ru BCT structures achieved through epitaxial growth are indeed ferromagnetic, and thereby closed the loop. For this work to not be considered as incremental, the authors need to provide more substantial new understanding. In its current form, I recommend the authors submit to a more specific journal.

Reply: Thank you very much for your review and comments. We hope we have addressed the concerns you have raised, and have replied to each of your comments, as denoted in red.

As for the scope of this work, we wish to point out the work by Shiki et al has limited characterization of the tetragonal structure, which does not satisfactorily prove the BCT metastable phase. And in addition, does not address in any form the magnetic properties of their claimed BCT structure in their work. We further would like to point out that it is not the creation of a metastable phase of Ru that is significant, but rather that showing the existence of a room temperature ferromagnetic element outside of Fe, Co and Ni that is significant. There have been several theoretical works predicting BCT Ru ferromagnetism, but it has never been demonstrated prior to our work! In addition, significant efforts to force other 4d elements (such as Pd) to achieve the Stoner criteria, necessary for ferromagnetism, have been reported on, but again the experimental demonstration is lacking. Given this context, our demonstration of the 4th room temperature ferromagnetic element of Ru is within the scope of Nature Communications. Finally, with respect to crystal structure, while this may seem minor, Shiki et al's work relies on a single crystal (110) Mo substrate which is prohibitively expensive for any actual application.

2. The authors demonstrated that it is possible to use Ru to create single element ferromagnetic materials at room temperature with a saturated magnetization of 148 emu/cm³, which can be increased to about 350 emu/cm³ after field annealing. These results are interesting and can provide addition insights for researchers in this field. To make the work more complete, I have the following suggestions:

In Fig. S2, the magnetic properties of epitaxial Mo/Al₂O₃ should also be shown as a reference and compared with that of the Ru/Mo/Al₂O₃ structures. The magnetic properties of Mo/Al₂O₃ are not clear in the manuscript. In addition, substrate heating during growth can be a potential source for introducing magnetic contaminations.

Reply: Thank you for this valuable comment, we have added the moment vs. field measurement for textured (110) Mo on the (110) Al₂O₃ substrate in Fig S3b. The measurement first shows that no ferromagnetism is caused from the Mo layer, and in addition since it was grown under identical in-situ heating conditions for the samples with Ru, this can rule out magnetic contamination arising from the heating process. Finally, we wish to point out that changes in the slope between the non-textured sample of sub\Mo(20)\Ru(2.5) and Mo control samples are more likely due to variations in the plastic mounts used in the VSM than due to the lack of Ru.

3. The saturated magnetization decreases with increasing the thickness of Ru. The authors argued that it is due to Ru relaxation into a non-ferromagnetic phase. More careful and strict microscopy studies may strengthen this claim, as current XRD data are not convincing. If the trend is true, more data points (varying thickness) are needed to show the critical thickness or turning point. Is there any possibility that the magnetism is resulting from Mo/Ru intermixing or interface coupling?

Reply: Thank you for your comment. We have added STEM images of the 2.5 nm and 12 nm Ru samples (Fig. S2 in revised manuscript). The STEM images support our additional XRD analysis of Fig 1c, which supports the claim of Ru relaxation (discussed on page 4). This closely ties in with our answer to your first comment. XRD and STEM point to regions of the samples having pure BCT Ru and other regions with a mixing of BCT and HCP phase Ru. In the 12 nm sample STEM (Fig. S2 of updated supplemental information), there is a clear region in which the BCT phase spans the entire film thickness. This supports our claim that the source of ferromagnetism does not come from an interfacial effect, but the electronic structure of tetragonal Ru. The interfacial strain serves purely to assist in nucleating tetragonal Ru growth.

With respect to the possibility of ferromagnetism coming from the Mo/Ru interface, we have seen no theoretical claims of this. Finally, we are planning future experiments utilizing polarized

neutron reflectivity and β -NMR to better understand the depth dependent profile of the magnetization since this provides information both about potential decreased M_s at the surface due to strain relaxation, and any interfacial effects.

4. If the BCT is stabilized by interfacial strain, and is the driving force for ferromagnetism, can the authors connect the strain differences due to thermal expansion at different T with changes in ferromagnetism?

Reply: Thank you for your comments. We would like to clarify that the ferromagnetism does not originate from the interfacial strain, but from the electronic structure of the BCT phase. This BCT phase can be obtained by proper epitaxy from a lattice matched seed layer which provides enough strain to nucleate this phase. This is clear based on STEM images of the 12 nm sample (Fig. S2 in supplemental), where regions of BCT Ru spanning the entire thickness exist—this is highly unlikely if it was purely an interfacial strain effect. We have not observed any significant changes in the magnetization as a function of temperature, which rules out the effect coming from a strain induced mismatched thermal coefficient.

5. For ferromagnetic materials, it is always useful to determine their Curie temperature. For the 2.5 nm Ru sample, 10 kOe and 100 °C were chosen as the annealing conditions. Is 100 °C above the Curie temperature? Magnetic field, T, and film thickness should be changed to provide more insights.

Reply: We do not know the Curie temperature of our sample, but we do know that it is above 400 K, since this is the limit of our measurement system. As we have mentioned in the major modifications section of this reply, the effect of field annealing has been removed from this manuscript, and will be followed up with in a future systematic study of field annealing effects.

7. It is difficult to visualize the changes in 2.5 nm field annealed sample. XRD and XRR results shown in Fig. S3 should be added to Fig. 1, overlapping with as-grown 2.5 nm sample.

Reply: Thank you for this suggestion, but since we have removed discussion surrounding field annealing this is no longer necessary. Please see reply to question 5.

8. Line 99, given the epitaxial relationship between Ru and Mo, the deviation should be estimated and compared to what predicted by calculation.

Reply: The difference between the predicted c/a and a values and our experimental results have been discussed in the paper, and so we assume this question is about the difference between the predicted Ru structure and (110) Mo. For our epitaxial relation the mismatch between (1-10) Mo and (011) Ru is ~3%; 5% between (001) Mo and (100) Ru. While this would be a large strain, we point out this is for theoretical epitaxy, and so not unreasonable. In addition, our STEM FFT shows that the measured Ru BCT structure has lattice constants closer to that of Mo.

The manuscript still contains many typos and mistakes:

1. MCA defined twice on the first page;
2. Line 54, UHV) should be UHV;
3. Line 71, RU should be Ru;
4. Line 108, “2.5 nm thick” was changed to “3 nm thick” in Figure description.
5. Line 124, which..., which...
6. Line 145, “cubic” should be “tetragonal”?

Reply: Thank you, we have corrected the typos you have found, and done additional proof reading to root out others.

Reviewer #3 (Remarks to the Author):

The authors claimed that they have developed a room temperature ferromagnetic metastable tetragonal phase of Ru, which indicates ferromagnetism at room temperature as the 4th single element ferromagnetic materials at room temperature outside of Fe, Co and Ni. The work is very interesting and important for magnetic and related community. However, there are some issues unclear, and they need more experiments for publishing these results in Nature communications.

Reply: Thank you very much for your review and comments. We hope we have addressed the concerns you have raised, and have replied to each of your comments, as denoted in red.

1, The magnetization measurement is a critical issue. I did not find the measurement method description in the text. Measured by SQUID? If yes, more magnetic measurements, such as magnetic spectrum or microscopic measurements, are needed to verify their claim.

Reply: The magnetization measurements were collected using VSM, which is discussed in the 'Methods' section, and we have clarified this in the main text as well. We have planned to collect XMCD and PNR on the samples, and beam time for such experiments is estimated to be in April of 2018 or later.

As another independent and insensitive method to any contamination from VSM measurement, we have carried out an anomalous Hall effect measurements on a textured and non-textured (sub\Mo(20)\Ru(6) deposited at room temperature, same as discussed in prior paragraph) samples (Fig. 5 in new manuscript). The textured 6 nm Ru sample shows ordinary Hall effect in addition to AHE, whereas the non-textured sample only shows the ordinary Hall effect. This result is shown in Fig. 5 and the related discussion is on pages 10-12. It is well known that AHE signal only responds to the perpendicular magnetization component of the sample, and since the samples in this work have no easy axis, a Hall resistance change is expected at the field necessary to overcome the demagnetization field of a thin film ($4\pi M_s$). This also helps rule out any contamination from ferromagnetic dust particles, since dust will not contribute to the conduction; if dust impurities in the target or from dust are the cause of ferromagnetism they would also show up in the control sample discussed.

We have also added additional control measurements by collecting M vs H curves for a sample with 20 nm of (110) Mo grown on the same (110) Al₂O₃ substrates, but no Ru. Table 1 was added in the supplemental information, which summarizes the number of samples made for each thickness, and the number of repeated M vs H ferromagnetic hysteresis loops for each sample. By repeating sample growth and ferromagnetic properties this also supports out claim.

2, The film of 2.5 nm was annealed and its magnetization was increased dramatically, how about the other thickness samples?

Reply: We are in the process of developing new samples to better understand the effect of field annealing with a more systematic study in the future. As such, we have removed discussion of field annealing from this work. As we mentioned in the major modifications section, since we do

not well understand this effect, we have removed it from the manuscript, and will follow up in a future study. In place of the removed discussion we have focused in on additional measurements to support our claim of ferromagnetic BCT Ru.

3, In the abstract, it seems that they prepared 2.5, 6, and 12 nm thick samples, but in Fig4, the samples are 3,6,12nm. Are they right?

Reply: The nominal (measured) thicknesses are 2.5 (2.7), 6 (6.6), and 12 (12.1) nm, this was confirmed by XRR. The thicknesses are denoted by nominal in the text, but all magnetization calculations are done using the measured thickness, and not nominal. Thank you for pointing out this typo.

4, There are some typos, For example, RU, should be Ru?, ferromagnetic metastable cubic phase of Ru, should be tetragonal?

Reply: Thank you for pointing out these errors, we have addressed them, and done additional proof reading to clean up all typos.

Reviewers' Comments:

Reviewer #1 (Remarks to the Author):

The manuscript is not sufficiently convincing even after revision.

The point is that the authors' claim is so important that the widely-accepted much more reliable way such as element-specific XMCD should be employed to directly deduce M-H curves of only Ru. This can be easily done by using XMCD even though the film contain magnetic contaminants such as Fe, Co and Ni.

Reviewer #2 (Remarks to the Author):

The authors have carefully and fully addressed the comments raised, and I recommend its publication in Nature Communications.

Reviewer #3 (Remarks to the Author):

I have no more questions since all my concerns were adressed by the authors. I agree to publish it.

Responses to Reviewers:

In order to make our case that XMCD is not required because we have already taken extensive steps to rule out the observed ferromagnetism is coming from Fe, Co or Ni impurities. There are four potential sources for magnetic contamination we have identified, which are discussed below along with the experiments we have conducted to rule this out.

1. The VSM and VSM sample holders:

This has been addressed by collecting moment vs. field measurements on the VSM sample holders both before and after all measurements reported in the main manuscript. An example of this is shown in Figure S3a, which clearly shows the VSM and sample holders only contribute a net paramagnetic signal, which can be trivially subtracted.

2. Impurities introduced from the targets used in sputtering deposition:

Impurities, such as a target poisoned by Fe, was a concern we have taken care to address since our group frequently works with Fe-based magnetic materials. This has been addressed by a non-textured, but with identical layer composition and thickness (substrate\Mo\Ru). The non-textured sample allows us to rule out that the ferromagnetism is not due to a poisoned target since any contaminants would be in the sample, regardless of in situ heating conditions and thin film texture. The non-textured sample shows a net diamagnetic signal (from the substrate) and no ferromagnetism. This is demonstrated from the XRD in Fig 1 and moment vs. field curve in Fig S3b. In addition to VSM methods, we have used transport measurements to look for ferromagnetism. We observe an anomalous Hall resistance in the textured Ru sample, but not the non-textured control sample—this supports our claims even further.

We have also collected STEM EDS, which has shown no discernable elements other than those predicted in our stack structure. Though EDS does have a large error in determining quantitative compositions, we can conclude there are no significant levels of Fe, Co or Ni.

3. The in-situ heating may introduce impurities:

In the first set of reviews by Nature Communications, it was suggested that the in-situ process could introduce contaminants to the samples, or that perhaps Mo might be the source of the ferromagnetism. This has been addressed by growing and measuring a (110) textured Mo sample under the same conditions as the substrate\Mo\Ru samples, but with no Ru layer. The moment vs. field curve shows no ferromagnetic behavior (Figure S3b). This simultaneously rules out the (110) Mo as the source of ferromagnetism, and possible introductions of impurities from the heating process during sputter deposition. Finally, we wish to add that in our system we primarily use transition metals for our samples, and the in-situ temperature of 400 C for heating is too low to evaporate these metals.

4. Magnetic dust on the sample once removed from the vacuum chamber:

Trace amounts of magnetic dust, or impurities, that may have ended up on the sample surfaces are the final concern, since magnetometers are quite sensitive to the magnetic moment. Transport measurements that show ferromagnetism were observed (Figure 5), and since dust on the sample

surface cannot contribute to the conduction, it is not possible this is the source of ferromagnetism.

Additionally, we have measured 12 total BCT Ru samples, all of which show similar M vs. H curves, with a total of 55 combined hysteresis loops. If dust was the source of contamination, it is highly unlikely the same magnetization would be so repeatable.

Reviewers' comments:

Reviewer #1 (Remarks to the Author):

Transport properties of ferromagnetic materials have been used to confirm whether the ferromagnetism originates from the material itself other than contaminants because it is so sensitive to magnetic moments in the film. Ferromagnetism in materials such as GaMnAs possessing a very small magnetic moment can be confirmed by this method like in Phy. Rev. B 57, R2037 (1998).

The point is that the anomalous Hall voltage (Hall resistance) versus H curve should be the same as the M-H curve by SQUID. Only one thing the reviewer still does not agree with is the external magnetic field applied was limited up to only 5kOe (not 10kOe) and there is no hysteresis in the R(Hall)- H curve. Why did the authors limit the H at 5 kOe ? It should be 10kOe.

This makes us easily compare them. If they are the same, there is no doubt about the occurrence of ferromagnetism in BCT Ru.

Responses to Reviewers:

Summary of changes:

We have revised the manuscript accordingly based on the latest comments from the reviewer #1. A list of revisions is shown as below. Those revisions are also highlighted in red in manuscript.

1. Added references related to the transport characterization, as is discussed in our response below.
2. Added discussion concerning the choice of the saturation field for R-H curves on page 12.
3. Added discussion about the differences in R-H and M-H curves in the similarly titled section of the supplementary information.
4. Miscellaneous formatting changes in accordance with the provided Nature Communications checklist.

NCOMMS-17-13038B

Reviewer #1 comments:

Transport properties of ferromagnetic materials have been used to confirm whether the ferromagnetism originates from the material itself other than contaminants because it is so sensitive to magnetic moments in the film. Ferromagnetism in materials such as GaMnAs possessing a very small magnetic moment can be confirmed by this method like in Phy. Rev. B 57, R2037 (1998).

The point is that the anomalous Hall voltage (Hall resistance) versus H curve should be the same as the M-H curve by SQUID. Only one thing the reviewer still does not agree with is the external magnetic field applied was limited up to only 5kOe (not 10kOe) and there is no hysteresis in the R(Hall)- H curve. Why did the authors limit the H at 5 kOe? It should be 10kOe.

This makes us easily compare them. If they are the same, there is no doubt about the occurrence of ferromagnetism in BCT Ru.

Response:

We would like to thank the reviewer very much for taking your time to review and respond to our latest revised manuscript. We are happy to respond your questions as follows.

First, in addition to the mentioned GaMnAs Phys. Rev. B paper referenced by the referee [1], we have also examined the work of Philip et al on ferromagnetic Cr doped In₂O₃ to help support our case that the measured Hall resistance carried out are enough for confirmation of ferromagnetism from BCT Ru. In this article, similar methods to those we utilize in our manuscript were used to determine the ferromagnetism in Cr doped In₂O₃ [2].

Second, in response to the field range measured, we have limited our field to only 5.5 kOe since this is enough field to overcome the demagnetization field from in plane to out of plane ($4\pi M_s$)

for the samples measured. The field range that we have swept is adequate to show both the anomalous Hall effect and the ordinary Hall effect of the Mo and Ru films. Since this field range covers the relevant ferromagnetic transport effects, and a larger field range simply shows more of the ordinary Hall effect dominated region. Due to these considerations, we have not collected higher field transport data.

A forward field sweep was only completed since the transport measurement is only to support our claim of ferromagnetism, and as an additional method to estimate the saturation magnetization. Since there is no perpendicular easy axis for all samples reported in our manuscript, no hysteresis in the AHE curve is expected.

Finally, we respectfully disagree with the statement that the $R(\text{Hall})$ vs H curve should be identical to the M - H curve. In the case of the resistance vs. field transport measurement, R_{Hall} is linearly dependent on M_z , which is directly proportional to H_z , and so R_{Hall} is expected to change linearly with H until the magnetization is saturated at $4\pi M_s$. In the case of M vs. H , however, the field is not expected to switch linearly from positive to negative saturation since we are not measuring along a hard anisotropy axis. Similar differences in M - H compared to $R(\text{Hall})$ - H were obtained in the article by Philip et al, mentioned previously. Shown below are the M - H and R_{Hall} - H measurements published by Philip et al (Figure R1), and in their results the curves do not have identical shape either. It should be noted that in the saturation region of the transport measurements by Philip et al there is a negative slope due to the ordinary Hall effect, which we also observed, but have subtracted out as a background signal from our measurements since it is simply an additional linear dependence. This negative slope is a useful confirmation that our results make sense, as our films are metallic and so the electrons are the majority carrier, which is confirmed in our BCT Ru samples and control samples.

Figure R1. M vs. H for Cr doped I_2O_3 films (left), and corresponding R_{Hall} vs field (left). Reprinted from Figures 3b and 4c from Philip et al, respectively [2].

In our measurements, the variation in the magnetization value obtained between the two measurements is due to any errors in estimating the magnetic volume of the Ru, whereas the

AHE method does not intrinsically rely on precisely knowing the volume. However, in both cases a reasonable, if smaller than expected, magnetization is found.

[1] F. Matsukura, H. Ohno, A. Shen, and Y. Sugawara. Transport properties and origin of ferromagnetism in (Ga,Mn)As. *Phys. Rev. B.* **57** R2037 (1998). Doi: <https://doi.org/10.1103/PhysRevB.57.R2037>

[2] J. Philip, A. Punnoose, B. I. Kim, K. m. Reddy, S. Layne, J. O. Holmes, B. Satpati, P.R. Leclair, T.S. Santos, and J. S. Moodera. Carrier-controlled ferromagnetism in transparent oxide semiconductors. *Nat. Mater.* Vol 4 (2006). <https://doi.org/10.1038/nmat1613>

Reviewers' Comments:

Reviewer #1 (Remarks to the Author):

The manuscript was well revised and now meets the standard of this journal. The reviewer still suggests the authors to perform XMCD measurement of Ru films in the near future.

Responses to Reviewers:

Comments from Reviewer #1:

The manuscript was well revised and now meets the standard of this journal. The reviewer still suggests the authors to perform XMCD measurement of Ru films in the near future.

Reply: Thank you for your kind review. We will carry out XMCD as soon as possible in addition to other more intrinsic methods of measuring the magnetization, such as PNR and beta-NMR.